# Aspirin Inhibition of Prostaglandin Synthesis Impairs Mosquito Egg Development

**DOI:** 10.3390/cells11244092

**Published:** 2022-12-16

**Authors:** Duyeol Choi, Md. Abdullah Al Baki, Shabbir Ahmed, Yonggyun Kim

**Affiliations:** Department of Plant Medicals, Andong National University, Andong 36729, Republic of Korea

**Keywords:** mosquito, prostaglandin, oogenesis

## Abstract

Several endocrine signals mediate mosquito egg development, including 20-hydroxyecdysone (20E). This study reports on prostaglandin E_2_ (PGE_2_) as an additional, but core, mediator of oogenesis in a human disease-vectoring mosquito, *Aedes albopictus*. Injection of aspirin (an inhibitor of cyclooxygenase (COX)) after blood-feeding (BF) inhibited oogenesis by preventing nurse cell dumping into a growing oocyte. The inhibitory effect was rescued by PGE_2_ addition. PGE_2_ was found to be rich in nurse cells and follicular epithelium after BF. RNA interference (RNAi) treatments of PG biosynthetic genes, including PLA_2_ and two COX-like peroxidases, prevented egg development. Interestingly, 20E treatment significantly increased the expressions of PG biosynthetic genes, while the RNAi of *Shade* (which is a 20E biosynthetic gene) expression prevented inducible expressions after BF. Furthermore, RNAi treatments of PGE_2_ receptor genes suppressed egg production, even under PGE_2_. These results suggest that a signaling pathway of BF-20E-PGE_2_ is required for early vitellogenesis in the mosquito.

## 1. Introduction

Mosquito-borne diseases are recognized as a leading killer of humans worldwide. The Asian tiger mosquito, *Aedes albopictus*, transmits at least 22 arboviruses, and it is a capable vector for transmitting Chikungunya, Dengue, and Zika viruses [1,2].

Female mosquitoes depend on blood-feeding (BF) to initiate egg development, which is mediated by several endocrine signals [3]. Insulin-like peptides can sense nutrient availability and stimulate the proliferation of germline stem cells [4]. The juvenile hormone (JH) acts as a main gonadotropic endocrine signal by activating vitellogenesis by the fat body to be competent to synthesize vitellogenin (Vg) in response to 20-hydroxyecdysone (20E) after BF [5]. Prostaglandins (PGs) have also been implicated in stimulating oviposition in some crickets, as PG-synthetic machinery is delivered from males during mating to produce massive amounts of PGs in female spermathecae [6]. The reproductive role of PGs is extended to mediate oogenesis in the fruit fly, *Drosophila melanogaster*, wherein PGE_2_ facilitates a nurse cell-dumping process to growing oocytes, in which a nurse cell cytoplasm is dumped into the oocyte [7]. PGE_2_ has also been shown to mediate eggshell protein synthesis during choriogenesis in several insects [8,9,10].

PGs are a group of oxygenated C20 polyunsaturated fatty acids that are typically derived from arachidonic acid (AA) [11]. AA is released from phospholipids by the catalytic activity of phospholipase A_2_ (PLA_2_), which has been identified in several insects [12,13]. AA is then oxygenated by cyclooxygenase (COX) to form PGs, by lipoxygenase to form leukotrienes (LTs), or by epoxygenase to form epoxyeicosatrienoic acids (EETs) [13]. In PG biosynthesis, insects use COX-like peroxidases, called peroxynectin [14,15] and heme peroxidase [16], which are likely to act in the same way as COX, because insect genomes do not contain any COX orthologs [17]. Specific PG synthases convert the COX product to PGE_2_ or PGD_2_, as seen in a lepidopteran insect, *Spodoptera exigua* [18,19]. In mosquitoes, PGE_2_ plays a role in mediating cellular and humoral immune responses [16,20]. However, little is known about the role of PG in mosquito reproduction.

In this study, we examine the effects of PG on *Ae. albopictus* oocyte development using aspirin, a potent anti-inflammatory drug that can inhibit PG biosynthesis [21]. The feeding or injection of aspirin to suppress PG production was shown to lead to a significant reduction in mosquito oocyte development. Furthermore, a crosstalk with 20E was assessed, and it showed that PG biosynthesis was dependent on 20E. Finally, the molecular action of PG signal in oocytes was further validated by the RNA interference (RNAi) specific to the PG receptor of *Ae. albopictus*.

## 2. Materials and Methods

### 2.1. Insect Rearing

*Ae. albopictus* was maintained in the following laboratory conditions: 27 °C temperature, 70 ± 10% relative humidity, and 16:8 h (L:D) photoperiod. First, instar larvae were reared in a plastic box (12 × 8 × 5 cm) containing 0.5~1.0 L of distilled water and supplemented with fish food mini pellets (Ilsung, Daegu, Republic of Korea). During the adult stages, 10% sucrose solution was supplied to adults using a cotton plug. Twice a week, female mosquitoes (4 days after adult emergence) were fed blood from an out-bred mouse line (SLC, Shizuoka, Japan) to initiate egg development. Each BF took 1 h in the afternoon (2~6 pm) at photophase. For females to oviposit, a wet kitchen towel was placed on a petri dish (8 cm in diameter) filled with distilled water.

### 2.2. Chemicals

Aspirin (ASP: 2-acetoxybenzoic acid), dexamethasone (DEX: (11β,16α)-9-fluoro-11,17,21-trihydroxy-16-methylpregna-1,4-diene-3), ibuprofen (IBU: α-methyl-4-isobutyl phenylacetic acid), prostaglandin E_2_ (PGE_2_: (5Z,11α,13E,15S-11,15-dihydroxy-9-oxoprosta-5,13-dienoic acid), naproxen (NAP: 6-methoxy-α-methyl-2-naphthaleneacetic acid), leukotriene B_4_ (LTB_4_: 5,12-dihydroxy-6,8,10,14-eicosatetraenoic acid), and 14,15-epoxyeicosatrienoic acid (14,15-EET) were purchased from Sigma-Aldrich, Korea (Seoul, Republic of Korea), and dissolved in dimethyl sulfoxide (DMSO, 10%). Antibody specific to PGE_2_ was purchased from Abcam (ab2318, Abcam, Cambridge, UK), and its secondary antibody conjugated with fluorescence isothiocyanate (FITC) was purchased from Thermo Fisher Scientific, Korea (Seoul, Republic of Korea).

### 2.3. Bioinformatics Analysis

DNA or amino acid sequences of different species were obtained from GenBank (www.ncbi.nlm.nih.gov (accessed on 12 December 2022)), with accession numbers mentioned in the figures. The phylogenetic tree was generated by the maximum likelihood tree method using software package MEGA-X, where evolutionary distances were computed using the Poisson correction method. Bootstrapping values were obtained as the results of 1000 repetitions to support branching and clustering. Protein domains were predicted using Prosite (http://prosite.expasy.org/ (accessed on 12 December 2022)), Pfam (http://pfam.xfam.org (accessed on 12 December 2022)), and InterPro (http://www.ebi.ac.uk/interpro/ (accessed on 12 December 2022)). N-terminal signal peptide was determined using SignalP 4.0 server (http://www.cbs.dtu.dk/services/SignalP/ (accessed on 12 December 2022)).

### 2.4. RNA Extraction and cDNA Preparation

Total RNAs were extracted from different developmental stages (~150 embryos, 50 young larvae, and one adult from male or female). To analyze adult body parts, 8-day-old adult females were used by isolating the head, thorax, ovary, and abdomen. To collect total RNA from the ovaries, 5-day-old females were allowed to feed blood. Total RNAs were extracted using Trizol reagent (Invitrogen, Carlsbad, CA, USA), according to the manufacturer’s instructions. After RNA extraction, RNA was resuspended in nuclease-free water and quantified using a spectrophotometer (NanoDrop, Thermo Fisher Scientific, Wilmington, DE, USA). RNA (500 ng) was used for cDNA synthesis with RT PreMix (Intron Biotechnology, Seoul, Republic of Korea) containing oligo dT primer, according to the manufacturer’s instructions.

### 2.5. RT-PCR and RT-qPCR

RT-PCR was performed using Taq DNA polymerase (GeneALL, Seoul, Republic of Korea) under the following conditions: initial denaturation at 94 °C for 5 min, followed by 35 cycles of 95 °C denaturation for 30 s, 55~57 °C for 30 s, 72 °C extension for 30 s, and a final extension at 72 °C for 10 min. Each RT-PCR reaction mixture (25 µL) consisted of cDNA template, dNTP (each 2.5 mM), 10 pmol for each forward and reverse primer (Appendix A), and Taq DNA polymerase (2.5 units/µL). The expression of a ribosomal protein S6 (RpS6, XM_019673337.2) was used as the reference gene.

All gene expression levels in this study were determined using a real-time PCR machine (Step One Plus Real-Time PCR System, Applied Biosystems, Singapore), while following the guidelines established by Bustin et al. [22]. Quantitative PCR (qPCR) was conducted with a reaction volume of 20 µL containing 10 µL of Power SYBR Green PCR Master Mix (Thermo Scientific Korea), 3 µL of cDNA template (200 ng), and 1 µL (10 pmol) of each of forward and reverse primers (Appendix A). After initial heat treatment at 95 °C for 2 min, qPCR was performed with 40 cycles of denaturation at 95 °C for 30 s, annealing at 55~57 °C for 30 s, and extension at 72 °C for 30 s. The expression level of RpS6 was used to normalize target gene expression levels under different treatments. Quantitative analysis was performed using the comparative CT (2^−ΔΔCT^) method [23]. All experiments were independently replicated three times.

### 2.6. Microinjection of Test Chemicals to Mosquito Females

For chemical treatment, females (1 day before BF) were first anesthetized on ice for 30 min. Next, these females were injected with 100 nL of test chemical with a glass capillary using a Sutter CO_2_ picopump injector (PV830, World Precision Instruments, Sarasota, FL, USA) under a stereomicroscope (SZX-ILLK200, Olympus, Tokyo, Japan). Prior to injection, microcapillaries (10 µL quartz, World Precision Instruments) with sharp points (<20 µm in diameter) were prepared with a Narishige magnetic glass microelectrode horizontal puller model PN30 (Tritech Research, Los Angeles, CA, USA).

### 2.7. ASP Injection and Rescue Experiment with PGE_2_ or Other Eicosanoids

To begin, 100 nL of ASP (1 μg/μL) was injected into each female. For the rescue experiment, another 100 nL of PGE_2_, 14,15-EET, or LTB_4_ (1 μg/μL) was injected into each female at 24 h after ASP treatment. Treated females were then blood-fed. After 3 days, whole ovaries were dissected in 100 mM phosphate-buffered saline (PBS, pH 7.4) under a stereomicroscope (Stemi SV11, Zeiss, Germany). Each treatment was replicated with 10 females.

### 2.8. Feeding Mosquito Females with ASP

Different concentrations of ASP were mixed with 10% sucrose solution and then provided to adult female *Ae. albopictus*. Sucrose (control) or ASP (1 mg/mL) in sucrose solution (5 mL) was placed in a vial with a cotton plug and provided to an adult cage containing 10 females 3 h before BF. Next, BF was performed using a mouse in a cage for 1 h. All mice were reared under pathogen-free conditions and anesthetized during BF. After BF, treated female mosquitoes were kept on the sucrose or sucrose-ASP solution for an additional 3 days prior to ovary dissection.

### 2.9. 20E and JH Injection into Female Mosquitoes without BF

The 5-day-old female mosquitoes were used for the injection of 20E and JH. First, 100 nL of JH (500 ng/µL) or 20E (500 ng/µL) was dissolved in 10% DMSO and then injected into females through the thorax. Treated female mosquitoes were kept on the sucrose solution. To analyze the time effects of JH and 20E, 50 ng of JH and 20E were injected. At 24 h post-injection, total RNA was extracted, as described above. Each treatment was replicated three times. Each replication was conducted with three adults.

### 2.10. Counting Oocytes at Different Developmental Stages

For oocyte counting, females were allowed to feed blood for 1 h in the afternoon (2–6 pm) at 5 days after adult emergence. Females were mated with males of the same age after BF and reared on a 10% sugar solution. The total number of oocytes was counted up to 4 days after BF.

### 2.11. Measurements of Mosquito Fecundity Following Aspirin Treatment

The oviposition rate was determined by observing the presence or absence of eggs at every 24 h interval from 72 h to 120 h after BF. At 120 h after BF, egg papers were fully submerged in water and allowed to hatch overnight. At 144 h after BF, the total number of laid eggs and the larval hatching rate were assessed.

### 2.12. Ovarian Developmental Analysis Using Fluorescence Microscopy

Ovaries from females (blood-fed) were dissected in 1 × PBS and fixed with 3.7% paraformaldehyde in a wet chamber under darkness at room temperature (RT) for 60 min. After washing three times with 1 × PBS, cells in ovarioles were permeabilized with 0.2% Triton X-100 in 1 × PBS at RT for 20 min. Cells were then washed another three times and blocked with 5% skim milk (MB cell, Seoul, Republic of Korea) in 1 × PBS at RT for 60 min. After washing another time, ovarian cells were incubated with FITC-tagged phalloidin in 1 × PBS at RT for 1 h. After washing another three times, cells were incubated with DAPI (1 mg/mL) diluted 1000 times in PBS at RT for 2 min for nucleus staining. After washing another three times, ovarian cells were finally observed under a fluorescence microscope (DM2500, Leica, Wetzlar, Germany) at 200× magnification.

To observe the PGE_2_ level, 1% PGE_2_ antibody was used in 1 × PBS after blocking with skim milk at RT for 2 h. After washing three times, 1% anti-rabbit-FITC conjugated antibody was used in 1 × PBS at RT for 1 h. After washing another three times, the nucleus was stained by DAPI (1 mg/mL) at RT for 2 min. The cells were then washed another three times and finally observed under a fluorescence microscope (DM2500, Leica, Wetzlar, Germany) at 200× magnification.

### 2.13. RNA Interference (RNAi)

Template DNA was amplified with gene-specific primers (Appendix A) containing T7 promoter sequence (5′-TAATACGACTCACTATAGGGAGA-3′) at the 5′ end. The resulting PCR product was used to in vitro synthesize double-stranded RNA (dsRNA) encoding *Ae. albopictus* genes using T7 RNA polymerase with NTP mixture at 37 °C for 3 h, according to the method outlined by Vatanparast et al. [24], using a MEGAscript RNAi kit (Ambion, Austin, TX, USA). dsRNA was mixed with a transfection reagent Metafectene PRO (Biontex, Plannegg, Germany) at a 1:1 (*v*/*v*) ratio and incubated at 25 °C for 30 min to form liposomes.

In the injection experiment, 300 ng of dsRNA was injected into 4-day-old females (1 day before BF) using a PV830 microinjector under an SZX-ILLK200 stereomicroscope. A control dsRNA (‘dsCON’) was prepared by the same method described above. To prepare dsCON, a 500 bp fragment of green fluorescence protein (GFP) gene was synthesized. After quantification using a nanodrop lite spectrophotometer, dsRNA was mixed with transfection reagent Metafectene PRO (Biontex, Plannegg, Germany) at a 1:1 (*v*/*v*) ratio. For each treatment, three females were mated with one male.

### 2.14. Ovary Measurement under RNAi Treatment Specific to PGE_2_ Biosynthetic Pathway-Related Genes

To begin, 100 nL of dsRNA (1 μg/μL) was injected into 4-day-old female mosquitoes 24 h before blood meal. After 48 h BF, female mosquitoes were anesthetized by being kept on ice for 30 min. The ovary was then dissected under a stereomicroscope (Nikon, Melville, NY, USA) and kept in PBS. For follicle and ovary size measurement, a stereomicroscope (M165FC, Leica, Germany) was used at 4× magnification.

### 2.15. Analysis of RNAi-Treated Females and PGE_2_ Rescue Experiment

For RNAi treatment, 300 ng of gene-specific dsRNA in 100 nL was injected into 4-day-old females at 24 h before BF. After BF, females were reared with 10% sugar solution. To rescue RNAi-treated females, 100 nL of PGE_2_ (1 μg) was injected into females at 24 h after dsRNA injection. Control RNAi (‘dsCON’) was injected at the same concentration. Females at 72 h after BF were used to count the total number of oocytes under a stereomicroscope (Stemi SV11, Zeiss, Germany). For the fecundity test, three females were mated with one male in each replication. Each treatment was replicated three times.

### 2.16. Statistical Analysis

All results are expressed as mean ± standard error. They were plotted using Sigma Plot (version 10.0, Systat Software, San Jose, CA, USA). Means were compared by the least square difference (LSD) test of one-way analysis of variance (ANOVA), conducted using PROC GLM of SAS program [25] and discriminated at Type I error = 0.05.

## 3. Results

### 3.1. Aspirin Suppressed Mosquito Egg Development

Newly emerged females of *Ae. albopictus* have previtellogenic oocytes (Figure 1A), where follicles develop soon after adult emergence. Each follicle contained several nuclei, and they increased in size with time after adult emergence. After BF, the ovary had vitellogenic oocytes and showed an increase in size. Each follicle was surrounded by the follicular epithelium and contained an oocyte and several nurse cells. The follicles grew slightly after adult emergence and rapidly increased in size soon after BF (Figure 1B). At 24 h after BF, the oocytes began to be enclosed with a chorion (Figure 1C). Finally, at 3 days post feeding, all oocytes were fully developed into chorionated oocytes, at which point, they were considered to be ready for oviposition.

To determine the role played by PG in mosquito oogenesis, different eicosanoid biosynthesis inhibitors were injected into females immediately prior to BF (Figure 2). Injection of dexamethasone (‘DEX’)—which serves as a general inhibitor of eicosanoid biosynthesis by inhibiting PLA_2_—impaired egg production. Injection of PG biosynthesis inhibitors (aspirin (‘ASP’) or ibuprofen (‘IBU’)) also inhibited egg production, whereas naproxen (‘NAP’), an LT biosynthesis inhibitor, did not (Figure 2A). The addition of PGE_2_ rescued the oogenesis of females treated with ASP (Figure 2B). The inhibitory activity of ASP on oogenesis led to a reduction in the number of laid eggs (Figure 2C). The reduced fecundity was rescued by the addition of PGE_2_, but not by other types of eicosanoids, such as 14,15-EET and LTB4. The eggs laid by the ASP-treated females also suffered from poor hatch rates (Figure 2D). The reduced egg hatch rate was significantly (*p* < 0.05) rescued by the addition of PGE_2_.

The inhibitory activities of ASP were assessed at different concentrations against oogenesis, with the results shown in Appendix A. As expected, ASP was found to inhibit the oogenesis of *Ae. albopictus* in a dose-dependent manner. In the injection of ASP into females, only 10 ng of ASP per female was required to reduce egg production (Appendix A). Moreover, at 1 μg of ASP, about 80% of the oocytes failed to produce eggs. The oral administration of ASP also inhibited oogenesis in a dose-dependent manner, although its inhibitory effect was much less than that of the injection method (Appendix A).

### 3.2. Aspirin Prevents Early Oogenesis by Inhibiting Nurse Cell Dumping

The inhibitory activity of ASP against oogenesis allowed us to monitor the presence of PGE_2_—a well-known PG in mosquitoes [16]—in the ovaries (Figure 3). The PGE_2_ signal was detected in the growing ovaries after BF (Figure 3A). Before BF, more than half of the volume of a follicle was filled with nurse cells (Figure 3B). BF stimulated growth in the oocyte size, along with a relative reduction in nurse cells. At 48 h after BF, there was only a tiny relic of nurse cells, which were presumably undergoing nurse cell dumping to the growing oocytes. During the nurse cell degeneration, the PGE_2_ signal was kept in both nurse cells and follicular epithelium in the growing follicles (Figure 3C). However, DEX or ASP treatment significantly (*p* < 0.05) reduced the PGE_2_ signal intensity. Under the reduced PGE_2_ signals, the oocytes did not grow in size, and the nurse cells were retained in the follicles.

### 3.3. PG Biosynthetic Genes of Ae. albopictus

The PGE_2_ signal suggested the presence of PG biosynthetic enzymes in *Ae. albopictus*. PLA_2_ catalyzes the first committed step of PG biosynthesis [12]. Six PLA_2_ genes were predicted from the genome of *Ae. albopictus*, and among the 16 PLA_2_ groups, two of these six genes were classified into Group III (*Aa-PLA_2_-IIIA* and *Aa-PLA_2_-IIIB*), while the other four were classified into Groups VI (*Aa-PLA_2_-VI*), VIII (*Aa-PLA_2_-VIII*), XII (*Aa-PLA_2_-XII*), and XV (*Aa-PLA_2_-XV*) (Appendix A). Five PLA_2_s were considered to likely be secretory or membrane-bound PLA_2_s, due to their signal peptides at N terminus (Appendix A). Among these five PLA_2_s, three PLA_2_s (*Aa-PLA_2_-IIIA*, *Aa-PLA_2_-IIIB*, and *Aa-PLA_2_-XII*) possess the Ca^2+^-binding domain. Meanwhile, Aa-PLA_2_VI has an active site that is categorized as a patatin-like lipase, suggesting a membrane-bound intracellular PLA_2_. These six PLA_2_s were expressed in all developmental stages of *Ae. albopictus* (Figure 4A). Four PLA_2_s were highly expressed in adults, and of these, *Aa-PLA_2_-XII* was the most frequent PLA_2_ transcript in female adults (Figure 4B). *Aa-PLA_2_-XII* was also highly expressed in the abdomens containing ovaries (Figure 4C). *Aa-PLA_2_-XII* expression was found to be highly up-regulated after BF (Figure 4D).

To synthesize PGs, PLA_2_ catalyzes phospholipids to release AA, which is then oxygenized by COX to produce PGH_2_, a common precursor of diverse PGs [26]. Based on the fact that a specific peroxidase, called peroxynectin (Pxt), performs COX-like catalytic activity in insects [20,27], peroxidase genes were annotated from the genome of *Ae. albopictus* and assessed by a clustering analysis with known COX-like genes in insects (Appendix A). Five peroxidases of *Ae. albopictus* were clustered with the four known COX-like Pxts (Appendix A). All these peroxidases possess a heme-binding domain (Appendix A). When comparing these peroxidase genes, in terms of their expressions after BF, only two peroxidases (*Aa-POX19* and *Aa-POX20*) were highly inducible to BF (Figure 4E). These results suggest that one PLA_2_ (*Aa-PLA_2_-XII*) and two peroxidases (*Aa-POX19* and *Aa-POX20*) are associated with PG biosynthesis in *Ae. albopictus*.

### 3.4. PGE_2_ Production Is Controlled by 20E after BF

The three PG biosynthetic genes were found to be highly inducible to BF. BF up-regulates 20E to initiate oogenesis in mosquitoes [28]. This suggests that 20E controls PG biosynthesis after BF. To test this hypothesis, 20E was assessed to control the expressions of PG biosynthetic genes (Figure 5). The 20E injection without BF significantly (*p* < 0.05) induced the expressions of the PG biosynthetic genes, while JH treatment did not (Figure 5A). The up-regulation of the genes was maintained at least 24 h after 20E injection. Moreover, 20E controlled the gene expressions in a dose-dependent manner (Figure 5B). However, a high (500 pg/adult) concentration of 20E was less active in inducing Aa-POX19 expression.

A loss-of-function approach using RNAi was applied to confirm the effect of 20E to control the expressions of PG biosynthetic genes. A 20E biosynthetic gene—*Aa-Shade*—was selected from the genome of *Ae. albopictus* and annotated to catalyze the conversion of ecdysone to 20E (Appendix A). Aa-Shade has the same sequence similarity as other dipteran insects (Appendix A), and it possesses functional domains, such as heme- or NADPH-binding domains typical to monooxygenases, that form 20E from ecdysone (Appendix A). Its expression in *Ae. albopictus* was inducible in BF (Figure 6A). However, the injection of dsRNA specific to Aa-Shade suppressed its inducible expression levels (Figure 6B). Under this RNAi condition, three PG biosynthetic genes were assessed in their expressions after BF (Figure 6C). The three PG biosynthetic genes were not induced in their expression levels, even after BF under the RNAi condition. However, the inducible expressions were rescued by the addition of 20E, while *Aa-Shade* expression was not.

### 3.5. Individual RNAi Treatments of 20E/PG Biosynthetic Genes Inhibit Oogenesis of Ae. albopictus

The 20E regulation of PG biosynthetic gene expressions suggested that 20E/PG biosynthetic genes are required for *Ae. albopictus* oogenesis. To test this hypothesis, individual RNAi treatments against these genes were performed through the injection of gene-specific dsRNAs, and these treatments resulted in significant reductions in their target gene expression levels (Appendix A). The RNAi treatments against target genes significantly inhibited oogenesis, while those against other non-target genes did not have such an effect on the oogenesis (Figure 7). For example, RNAi of *Aa-Shade* significantly suppressed oogenesis (Figure 7A). Among six PLA_2_s of *Ae. albopictus*, four PLA_2_s were expressed in adults, and their RNAi treatments were compared, in terms of suppressing oogenesis. Only two RNAi treatments specific to *Aa-PLA_2_-IIIB* or *Aa-PLA_2_-XII* significantly (*p* < 0.05) suppressed oogenesis, where the RNAi of *Aa-PLA_2_-XII* was more effective than Aa-PLA_2_IIIB in suppressing oogenesis (Figure 7B). Either RNAi specific to *Aa-POX19* or *Aa-POX20* significantly suppressed the oogenesis. However, RNAi specific to non-PG biosynthetic peroxidase (*Aa-POX15*) did not have any adverse effect on the oogenesis.

### 3.6. RNAi of PGE_2_ Receptor (PGE_2_R) Expression Inhibits Oogenesis

The stimulating effect of PGE_2_ on oogenesis suggests that there is a PGE_2_R on the follicles of *Ae. albopictus*. The interrogation of the *Ae. albopictus* genome with the *M. sexta* PGE_2_ receptor sequence [29] allowed us to predict a PGE_2_R gene (Aa-PGE2R). *Aa-PGE_2_R* encoded 398 amino acid residues and 7 transmembrane domains (Appendix A). Its predicted amino acid sequence was clustered with other vertebrate EP4 receptors (Appendix A). Aa-PGE_2_R was expressed in all developmental stages (Appendix A). It was highly expressed in adult females (Appendix A). In female adults, it was shown to be highly expressed in the ovaries. The expression of *Aa-PGE_2_R* in the ovaries increased with egg production (*r* = 0.9972; *p* < 0.0001) (Appendix A). dsRNA specific to *Aa-PGE_2_R* significantly (*p* < 0.05) suppressed its expression level when it was injected into teneral female adults (Figure 8A). Compared to control females, RNAi-treated female adults failed to develop egg production. PGE_2_ addition to RNAi-treated females did not rescue the reduction in egg production.

### 3.7. No Effect of Aspirin on Vitellogenin (Vg) Expression

Two Vg genes were predicted from *Ae. albopictus* genome: *Aa-Vg1* and *Aa-Vg2* (Appendix A). Their expressions were highly inducible to BF, wherein the *Aa-Vg1* transcript level was much higher than that of *Aa-Vg2* (Appendix A). The 20E injection significantly induced both Vg genes. However, ASP injection did not inhibit the up-regulation of the two Vg gene expressions, which were induced by 20E (Figure 8B).

## 4. Discussion

Most human disease-transmitting mosquitoes are anautogenous and depend on blood meal for their reproduction, especially to produce egg proteins. In protein synthesis, four gonadotropic signals—such as to JH, 20E, insulin-like peptide (ILP), and amino acids—are well-known to solely or cooperatively mediate the Vg gene expression to promote vitellogenesis in mosquitoes [30]. However, other oogenesis processes, such as meroistic oocyte development and choriogenesis in mosquitoes, are not well-understood. Mosquito ovarioles are polytrophic, and each follicle consists of oocyte, nurse cells, and follicular epithelium, as can be seen in Figure 1. During vitellogenesis, oocytes increased in size with the accumulation of Vg proteins and other maternally derived materials, while the nurse cells became relatively smaller and then lost. Here, we did not know which endocrine signal (or signals) mediates (or mediate) this follicle development. This study reports on the role of PGE_2_ in mediating oocyte maturation through nurse cell dumping in *Ae. albopictus*.

ASP treatment to inhibit PG biosynthesis inhibited the oogenesis of *Ae. albopictus*. However, naproxen treatment to inhibit LT biosynthesis did not inhibit mosquito oogenesis. The addition of PGE_2_ to the ASP-treated mosquitoes rescued the egg production, while the addition of other eicosanoids did not. Among PGs, PGE_2_ was detected in mosquitoes because specific stellate cells of Malpighian tubules possessed PGE_2_ in *Ae. aegypti* to mediate fluid secretion [31]. PGE_2_ was produced in the hemolymph of *An. gambiae*, and it played a crucial role in defending from *Plasmodium* infection [16]. In *An. albimanus*, PGE_2_ was detected in the midgut, and it mediated the gene expression of antimicrobial peptides [20]. These results suggest that PGE_2_ might be produced and mediate oogenesis in mosquitoes in our current study. Indeed, PGE_2_ was observed in the nurse cells and follicular epithelium in the growing follicles of *Ae. albopictus*. Nurse cell dumping is required for oocyte development in insects containing polytrophic ovarioles [32]. In *Drosophila*, such nurse cell dumping is mediated by PGs via cytoskeleton rearrangement by bundling actin filament through Fascin [7]. In *S. exigua*, PGE_2_ stimulates a small GTPase, Cdc42, to activate Fascin by binding to its specific PGE_2_ receptor [33]. Later, the PGE_2_ binding to its receptor has been shown to trigger cAMP release and the subsequent up-regulation of Ca^2+^, which activates small G proteins to stimulate actin polymerization and bundling to modulate cytoskeletal rearrangement [34]. The inhibition of PG biosynthesis prevented the oogenesis of *S. exigua* because nurse cell dumping did not occur, as was the case in our current study using mosquitoes [16]. This suggests that PGE_2_ induces the Ca^2+^ signal, which activates the cytoskeletal rearrangement of nurse cells to dump their contents into the growing oocytes in *Ae. albopictus*.

We identified the PGE_2_ receptor of *Aa. albopictus* and demonstrated its role in mediating oogenesis in two mosquito species. Unlike the lepidopteran PGE_2_ receptors of *M. sexta* and *S. exigua*, which are classified into the EP2 type [29,35], the predicted amino acid sequence of *Aa-PGE_2_R* shared a homology with EP4 among four identified EP receptors. EP4 has been implicated in various physiological and pathological responses in mammals [36]. When bound to PGE_2_, EP4 can mobilize trimeric G proteins to be dissociated into Gαs and Gβγ components to stimulate adenyl cyclase and increase the intracellular levels of cAMP. EP4 activation of G proteins can also activate the PI3K/AKT/mTOR, ERK, and p38 MAPK pathways [37]. These findings suggest that Aa-PGE_2_R can modulate various physiological processes using these down-stream signal pathways in *Ae. albopictus*. For example, *Aa-PGE_2_R* was highly expressed in both the larval and adult stages. This suggests that Aa-PGE_2_R can mediate physiological processes other than adult reproduction. Actin cytoskeletal rearrangement in the hemocytes of *S. exigua* is mediated by PGE_2_ during hemocyte-spreading to perform cellular immune responses [18]. This suggests that Aa-PGE_2_R may mediate hemocyte behavior, as well as oogenesis, in *Ae. albopictus*.

The PG signal mediating mosquito oogenesis was induced by 20E after blood meal in *Ae. albopictus*. To synthesize PGs, two initial catalysis steps are carried out, first by PLA_2_ and subsequently by COX in mammals [26]. This study predicted six PLA_2_ genes from *Ae. albopictus* genome, of which *Aa-PLA_2_XII* was highly expressed in adult females and induced its expression after BF. Moreover, its RNAi led to a significant reduction in the egg production of *Ae. albopictus*. These results suggest that Aa-PLA_2_XII is associated with *Ae. albopictus* oogenesis, likely by producing a precursor of PG biosynthesis. COX is not encoded in mosquitoes as well as it is in other insects, except for the human body louse *Pediculus humanus corporis* [17]. Instead, insects are likely to use COX-like peroxynectin (Pxt) [14]. Two Pxts have been identified in *S. exigua* and in the mediation of PG biosynthesis [15]. Similar peroxidases are found in another mosquito, *An. gambiae* [16]. Based on these COX-like genes, this current study predicted two peroxidases (*Aa-POX19* and *Aa-POX20*) to be COX-like genes. These two POX genes were inducible to BF, and their RNAi treatments inhibited egg production. These results suggest that Aa-PLA_2_XII and two POXs sequentially catalyze PG biosynthesis to stimulate *Ae. albopictus* oogenesis. In mosquitoes, oogenesis consists of two phases, i.e., previtellogenesis after adult eclosion and vitellogenesis after BF, and these are mainly regulated by JH and 20E, respectively [28]. JH is typically released soon after adult emergence, and it mediates the differentiation of fat body to be competent to synthesize vitellogenin in response to 20E by increasing ribosomal biogenesis [38]. During previtellogenesis, JH down-regulates the expressions of two developmental transcriptional factors (Krüppel homolog 1 and Hairy), while allowing the mosquitoes to acquire responsiveness to 20E after BF by mediating the translation of the competence factor βFTZ-F1 (fushi tarazu binding factor 1) [39,40]. After a blood meal, a female mosquito can synthesize Vg in response to 20E [41]. JH again activates follicular patency to facilitate Vg uptake by growing oocytes [5]. During vitellogenesis, 20E up-regulates ecdysone-induced protein 74 (E74), ecdysone-induced protein 75 (E75), and Broad, which act together to express Vg gene [42,43]. At the terminal phase of vitellogenesis, ecdysone-induced protein 93 (E93) induces Hormone Receptor 3 (HR3) to shutdown Vg expression and βFTZ-F1 activation to inhibit the target of rapamycin (TOR) signaling and activate programmed autophagy [44]. In our current study, using *Ae. albopictus*, 20E is found to activate PG signaling by up-regulating Aa-PLA_2_XII and two POX gene expressions to mediate the nurse cell dumping of the vitellogenic follicles. The association of 20E with PG biosynthesis was supported by two different treatments. Without BF, 20E alone induced the expressions of PG biosynthetic genes. Conversely, after BF, PG biosynthesis was blocked by the RNAi treatment of Aa-Shade, a 20E biosynthetic gene. However, the PG signaling was not associated with Vg expression in *Ae. albopictus*. Vg expression was induced by 20E after BF. Collectively, our results suggest that BF stimulates 20E-mediated vitellogenesis by activating Vg synthesis under 20E signal and independently facilitating nurse cell dumping by the PG signaling pathway activated by 20E (Figure 9).

Our results propose a novel endocrine signal that uses PGs in mosquito oogenesis. In insects, the first report on the reproductive role of PG concerned the egg-laying behavior of house cricket (*Acheta domesticus*) females [45]. Loher [46] analyzed PGE_2_ levels in females and found a significant increase in hormone levels after mating. The PGE_2_ levels in mated females are much higher than those in the male reproductive organ. This led to a hypothesis of PG-synthetic machinery transfer during mating, wherein males transfer PGE_2_-biosynthetic enzymes in the spermatophores, and the mated females subsequently synthesize PGE_2_ in the spermathecae to release PGE_2_ to the hemolymph. Moreover, Loher et al. [6] showed that cricket spermatophores exhibited PGE_2_ biosynthetic activity that was specifically inhibited by aspirin. In addition to the egg-laying behavior, PGs mediate vitellogenesis by stimulating nurse cell dumping in insect reproduction. The finding of this PG signal expands our understanding of mosquito oogenesis, as modulated by JH, 20E, and ILP. The inhibition of egg development by ASP also holds great promise as a novel control strategy against important mosquito vector species. In the current study, orally administered ASP interfered with egg development, thus demonstrating its potential application in reducing mosquito populations. Moreover, PGs are likely to play crucial roles in the gut immunity of mosquitoes against microbiota [16]. The inhibition of PG production by ASP treatment might also impair mosquito immunity, thus making mosquitoes more susceptible to microbial pathogens. Therefore, designing and synthesizing ASP derivatives might have potential in the development of highly potent mosquitocidal agents.

## Figures and Tables

**Figure 1 cells-11-04092-f001:**
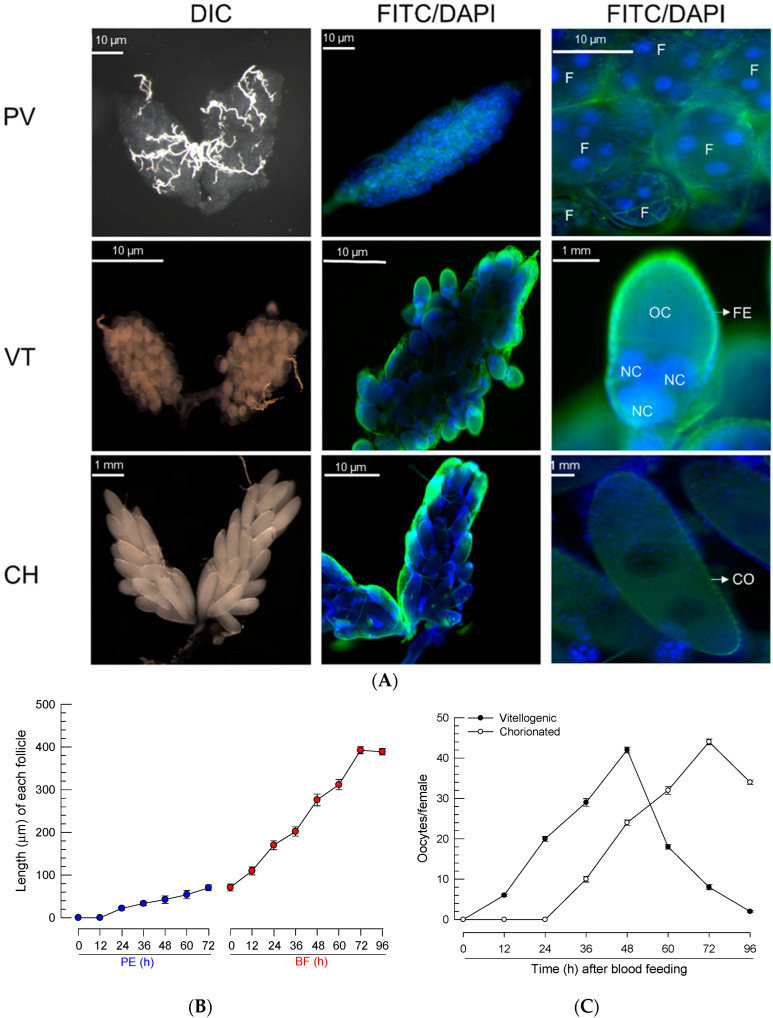
Oocyte development of *Ae. albopictus* females. (**A**) Three developmental stages of oocytes: previtellogenic (‘PV’), vitellogenic (‘VT’), and chorionated (‘CH’). Ovaries were observed at differential interference contrasts (‘DIC’). Their cells were stained with FITC-tagged phalloidin (green) and DAPI (blue). Follicles (‘F’) were visible at the PV stage. At the VT stage, oocyte (‘OC’) and nurse cell (‘NC’) can be seen to be enclosed with follicular epithelium (‘FE’). At the CH stage, the oocyte can be seen to be chorionated (‘CO’). (**B**) Follicle growth in ovaries post-adult emergence (‘PE’) or post-blood feeding (‘BF’). (**C**) Oocyte development. Total ‘VT’ and ‘CH’ oocytes were counted under a stereomicroscope at the selected time points, up to 96 h post-BF. For each treatment, 10 females were used to analyze follicle growth and oocyte development.

**Figure 2 cells-11-04092-f002:**
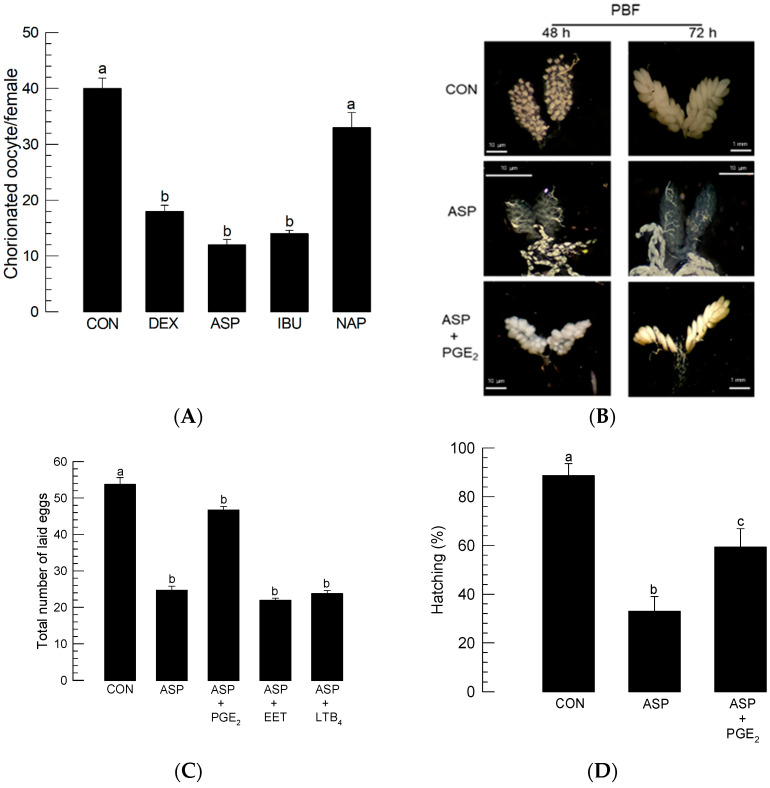
Effects of different eicosanoid biosynthesis inhibitors on the egg production of *Ae. albopictus*. PLA_2_ inhibitor (dexamethasone: ‘DEX’), two COX inhibitors (ibuprofen: ‘IBU’, aspirin: ‘ASP’), and LOX inhibitor (naproxen: ‘NAP’) were used to treat 5-day-old females before blood-feeding (‘BF’). At 10 min before BF, the inhibitor was injected into females at a dose of 1 μg/individual. Dimethyl sulfoxide (‘DMSO’, 10%) used to dissolve inhibitors was injected as a control (‘CON’). (**A**) Effect of inhibitors on the number of chorionated oocytes at 72 h after BF (‘PBF’). Each treatment was assessed using 10 females. (**B**) Rescue effect of PGE_2_ (100 ng/adult) on choriogenesis of females treated with ASP (100 ng/adult). Ovaries were observed at 48 h and 72 h PBF. (**C**) Rescue effect of PGE_2_ on choriogenesis of ASP-treated females. ASP was injected at a dose of 1 μg/individual. PGE_2_, 14,15-EET, or LTB4 was injected into ASP-treated females at a dose of 100 ng/adult. After BF, three females were allowed to mate with one male in each replication. Each treatment was replicated three times. The number of laid eggs was counted for 6 days PBF. (**D**) Decrease in the hatching rate of eggs laid by the ASP (1 µg/individual)-treated females. To capture the rescue effect, PGE_2_ (100 ng/adult) was injected along with ASP. Each replication used 100 eggs. Each treatment was replicated three times. The different letters above the standard error bars indicate significant differences among means at Type I error = 0.05 (LSD test).

**Figure 3 cells-11-04092-f003:**
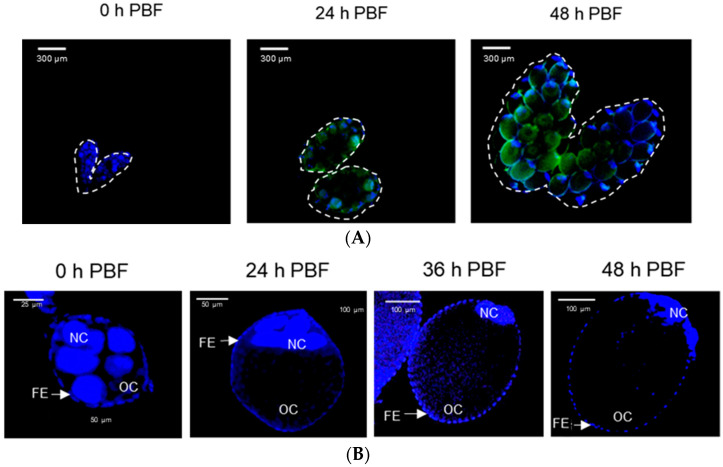
Detection of PGE_2_ signals in the growing follicles of *Ae. albopictus*. (**A**) Oogenesis after blood-feeding (‘PBF’), along with PGE_2_ signals (green colored) in the ovary. (**B**) Nurse cell dumping during oogenesis. ‘OC’, ‘FE’, and ‘NC’ stand for oocyte, follicular epithelium, and nurse cell, respectively. (**C**) Inhibitory effects of eicosanoid biosynthesis inhibitors on the induction of PGE_2_ signal and subsequent prevention of nurse cell dumping. Dexamethasone (‘DEX’) or aspirin (‘ASP’) was injected at a dose of 100 ng/individual. Dimethylsulfoxide (‘DMSO’, 10%) used to dissolve inhibitors was injected as a control (‘CON’). PGE_2_ fluorescence intensity levels were quantitatively measured at the oocytes (*n* = 20) after inhibitor treatments using ImageJ software (https://imagej.nih.gov/ij (accessed on 12 December 2022)). Different letters above standard error bars indicate significant differences among means at Type I error = 0.05 (LSD test).

**Figure 4 cells-11-04092-f004:**
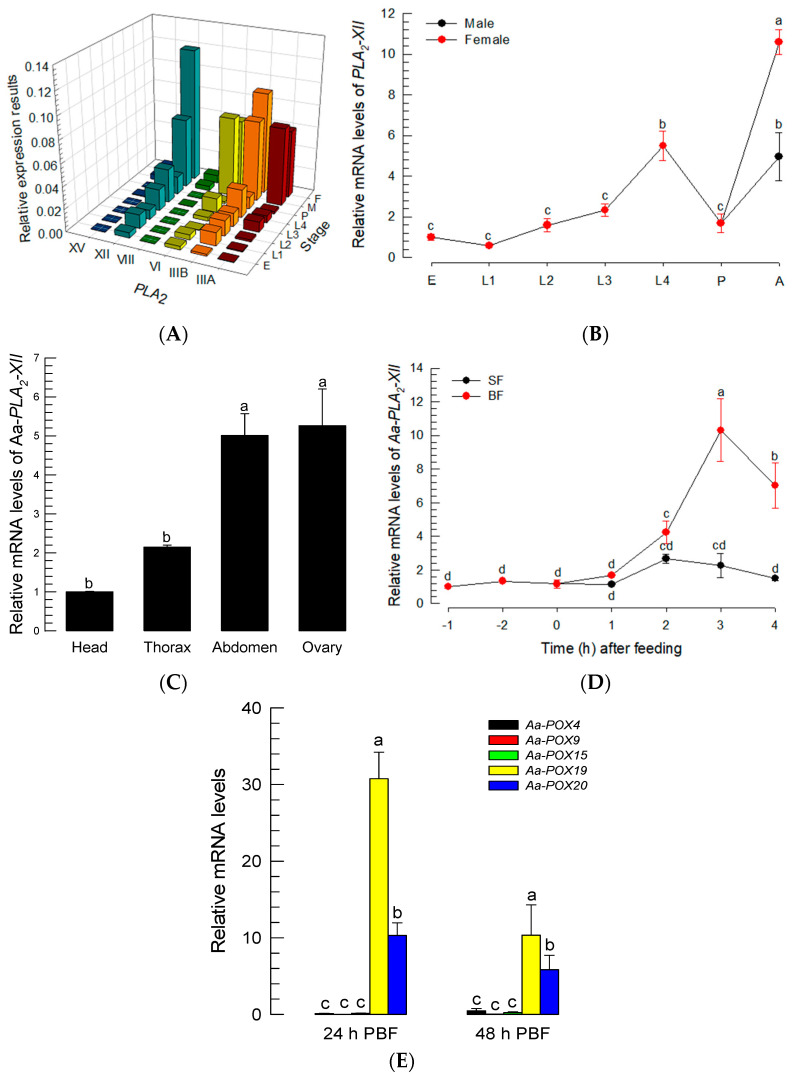
Prediction of PLA_2_ and COX-like POX genes in PG biosynthesis of *Ae. albopictus*. (**A**) Expression profiles of six Aa-PLA_2_s at different developmental stages: ‘E’ for egg, ‘L1-L4′ for 1st-4th instar larvae, ‘P’ for pupa, ‘M’ for male, and ‘F’ for female. Expression levels of *Aa-PLA_2_-IIIA* (‘IIIA’), *Aa-PLA_2_-IIIB* (‘IIIB’), *Aa-PLA_2_-VI* (‘VI’), *Aa-PLA_2_-VIII* (‘VIII’), *Aa-PLA_2_-XII* (‘XII’), and *Aa-PLA_2_-XV* (‘XV’) were analyzed by RT-qPCR. (**B**) Expression profiles of *Aa-PLA_2_-XII* in male and female mosquitoes. ‘A’ stands for adults. (**C**) Expression profile of *Aa-PLA_2_-XII* in the different tissues of adult female mosquitoes. (**D**) Effects of sugar-feeding (‘SF’) and blood-feeding (‘BF’) on the expression of *Aa-PLA_2_-XII*. (**E**) Expression profiles of five *Aa-POX* genes PBF. Each treatment was replicated three times with independent tissue preparations. The different letters above the standard error bars indicate significant differences among means at Type I error = 0.05 (LSD test).

**Figure 5 cells-11-04092-f005:**
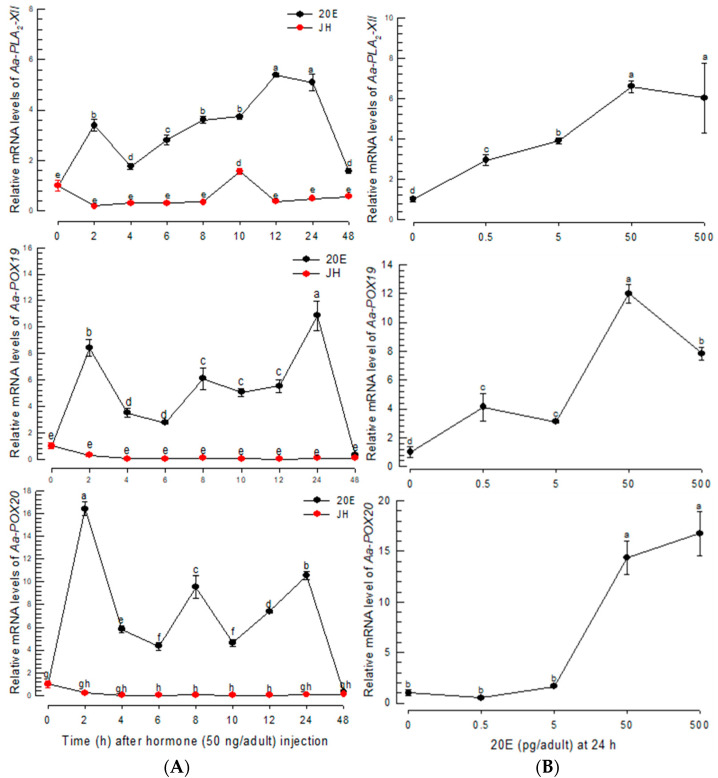
Inducible expressions of PG biosynthesis-associated genes in response to 20E in *Ae. albopictus*. (**A**) Expression profiles of *Aa-PLA_2_-XII*, *Aa-POX19*, and *Aa-POX20* in response to 20E (50 ng/adult) and JH (50 ng/adult). (**B**) Dose-response of 20E against the gene expressions. Dimethylsulfoxide (‘DMSO’, 10%) used to dissolve hormones was injected as a control (‘CON’). Three insects were used for each treatment. Each treatment was replicated three times with independent tissue preparations. The different letters above the standard error bars indicate significant differences among means at Type I error = 0.05 (LSD test).

**Figure 6 cells-11-04092-f006:**
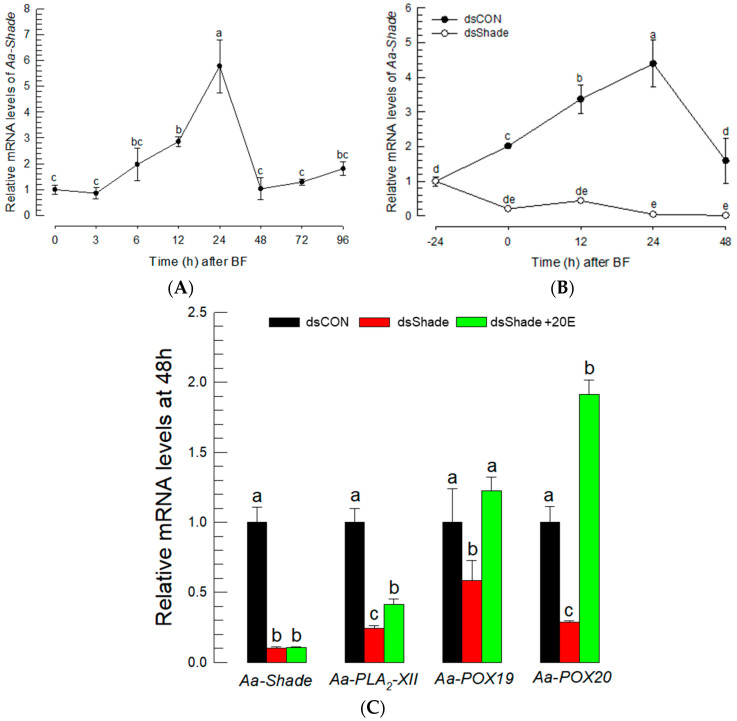
Suppression of PG biosynthesis-associated gene expressions by RNAi specific to a 20E biosynthetic gene, *Aa-Shade*, in *Ae. albopictus*. (**A**) Expression profile of *Aa-Shade* after blood-feeding (‘BF’). (**B**) Suppression of Aa-Shade expression by injecting its specific dsRNA (‘dsShade’, 300 ng/adult). (**C**) Suppression of the induced expressions of *Aa-Shade*, *Aa-PLA_2_-XII*, *Aa-POX19*, and *Aa-POX20* after BF by RNAi specific to Aa-Shade, and rescue by the addition of 20E. A *GFP* gene was used as a control dsRNA (‘dsCON’). Each treatment was replicated three times. The different letters above the standard error bars indicate significant differences among means at Type I error = 0.05 (LSD test).

**Figure 7 cells-11-04092-f007:**
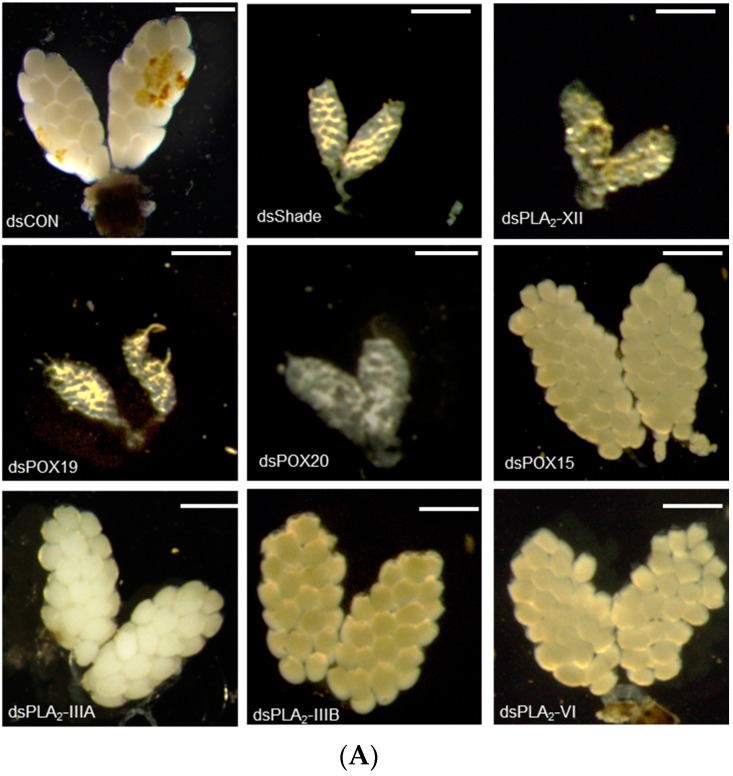
Functional association of PG biosynthesis-associated genes on the ovary development in *Ae. albopictus*. (**A**) Effect of individual RNAi on ovary development. The RNAi treatments were specifically targeted against *Aa-Shade*, *Aa-PLA_2_-XII*, *Aa-PLA_2_-IIIA*, *Aa-PLA_2_-IIIB*, *Aa-PLA_2_-VI*, *Aa-POX19*, *Aa-POX20*, or *Aa-POX15*. dsRNA (300 ng/adult) was injected into 4-day-old female mosquitoes before 24 h blood feeding (‘BF’). At 48 h after BF, the ovary development was assessed. Each scale bar indicates 400 μm. A *GFP* gene was used as a control dsRNA (‘dsCON’). (**B**) Comparison of ovary and its follicle developments among RNAi treatments. Each treatment was replicated three times, with three randomly chosen mosquitoes. The different letters above the standard error bars indicate significant differences among means at Type I error = 0.05 (LSD test).

**Figure 8 cells-11-04092-f008:**
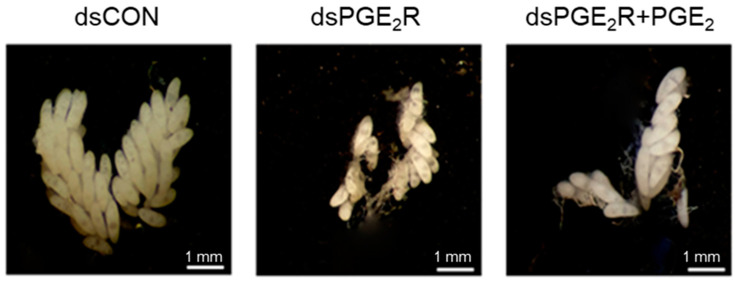
PGE_2_ mediates oogenesis through its specific receptor (Aa-PGE_2_R), but it does not induce Vg production. (**A**) Influence of RNAi specific to *Aa-PGE_2_R* on choriogenesis. RNAi was performed by injecting 300 ng of gene-specific dsRNA (‘dsPGE_2_R’) or control dsRNA (‘dsCON’) into 5-day-old females at 10 min before blood-feeding (BF). Changes in mRNA levels were assessed by RT-qPCR at 72 h after injecting dsPGE_2_R or dsCON. To confirm the RNAi treatment, PGE_2_ (100 ng per female) was injected and accompanied by dsRNA (300 ng/adult) treatment. At 72 h after BF, the total number of chorionated oocytes was counted under a stereomicroscope. Each treatment used 10 females. (**B**) Effect of ASP (100 ng/adult) on expressions of two vitellogenin genes (*Aa-Vg1* and *Aa-Vg2*) in response to 20-hydroxyecdysone (20E, 50 ng/adult) injection. Treated females were not blood-fed. ‘ASP+20E’ represents the co-injection of two chemicals. All treatments were independently replicated three times. The Vg gene expression levels were measured by RT-qPCR at 24 h after the chemical treatments. The different letters above the standard error bars indicate significant differences among means at Type I error = 0.05 (LSD test).

**Figure 9 cells-11-04092-f009:**
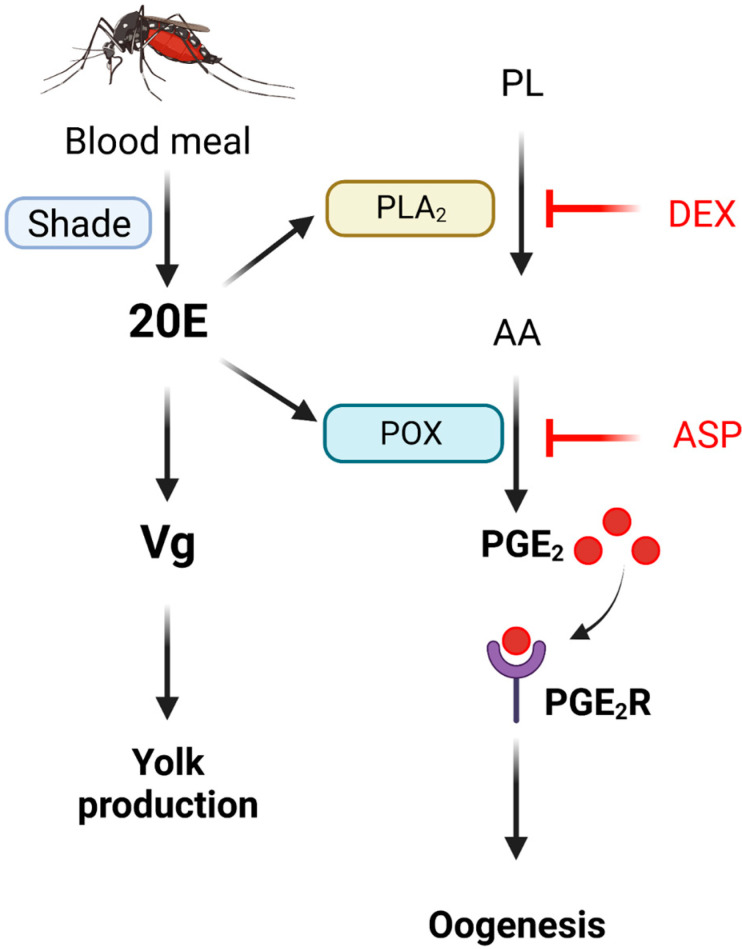
Working hypothesis of PGE_2_ on oogenesis of *Ae. albopictus*. Oogenesis is stimulated by blood-feeding via an endocrine signal of 20-hydroxyecdysone (‘20E’), which is synthesized by the catalytic activity of a cytochrome P450 monooxygenase called ‘Shade’. The 20E stimulates vitellogenin (‘Vg’) synthesis in fat body to be accumulated in the growing oocytes. The 20E also stimulates PGE_2_ biosynthesis in the follicles by inducing the gene expressions of ‘PLA_2_′ and COX-like ‘POX’. PLA_2_ catalyzes the release of arachidonic acid (‘AA’) from phospholipid (‘PL’). This reaction is specifically inhibited by dexamethasone (‘DEX’) treatment. COX-like POX catalyzes the oxygenation of AA to form PGE_2_. This is specifically inhibited by aspirin (‘ASP’) treatment. The formed PGE_2_ then binds to its specific receptor (‘PGE_2_R’) on the ovarian membranes and mediates oogenesis.

## Data Availability

Not applicable.

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
