# Peer review of "Aspirin Inhibition of Prostaglandin Synthesis Impairs Mosquito Egg Development"

_cells, 2022, doi:10.3390/cells11244092_

Round 1
Reviewer 1 Report
The hormonal triggers of egg development in mosquitoes following a blood meal have been a topic of research for decades given their importance to reproduction and persistence of these disease vectors. The authors have identified a previously unidentified endocrine signal triggering egg development, involving mosquito prostaglandins. The authors have conducted a detailed characterisation of this signal pathway, including identifying key signals and receptors, with evidence provided by robust gene expression changes of key genes and phenotypic changes, shown through morphometric analysis and exquisite immunohistochemistry of mosquito oocytes. The experiments are thorough and well executed.
Main comments
Page 1, para 2. It is not clear what is meant by the term “cell-dumping”. Please explain this term at when it is first used.
Page 1, para 3. The statement is made that insect genomes do not contain any COX ortholog. However, the second sentence of the paragraph mentions insects and the third paragraph discusses the action of COX. This is confusing as it suggests that COX proteins are found in insects. Furthermore, later in the paragraph states “specific PG syntases convert the COX product..” It is unclear how COX could be playing a role in insects if insects dot contain any COX orthology. Later, in the Discussion, the statement is made that COX proteins occur in vertebrates and not insects. This should be stated here in the Introduction.
Page 3, paragaphs 2 and 3. The section on “RT-PCR and RT-qPCR” is highly confusing. The authors refer to “real time PCR” and it seem that they abbreviate this as RT-PCR. However, the abbreviation “RT-PCR” should be used to refer to “Reverse Transcriptase PCR”. It does not seem that the authors mean Reverse-transcriptase PCR in the first sentence of paragraph 2, as the sentence mentions the use of taq polymerase. Revese trancriptase PCR requires the use of a reverse transcriptase, but only taq is mentioned here. Istrongly suggest that the authors do not use the term “Real time PCR” but instead use “quantitative PCR”. It is not clear how the methods described in paragraph 3 differ from those in paragraph 2 (it seems some of the Methods are repeated?).
Minor comments
Page 1, first paragraph. Mosquitoes do not “cause” viruses. I suggest the words “and it is a
capable vector causing” are replaced with “including”
Page 1, paragraph 2. Incorrect term “vitellogenesis of fat body”. Should this be “vitellogenesis by the fat body”
Page 1, pare. 2. Change “spermathecal” to “spermathecal”
Page 1, para 2. Change “in a fruit fly” to “in the fruit fly”
Page 2, para 4. What was the final concentration of DMSO in the treatments?
Page 3, para 4. Should “sutter” be capitalised? Is this a proper noun?
Page 3, paras 5 and 6. The first part of Section 2.7 seems to repeat what is said in Section 2.6.
Page 3, para 7. What was the strain of mouse? State how the mice were anaesthetised.
Page 4, para 6 (section 2.13). Should “before 24 h bloodmeal” be “24 h before a blood meal”?
Page 4, para 7 (section 2.14). Does this section describe the double stranded RNA constructs that were used in section 2.13? If so, I suggest that this information be placed before section 2.13, possibly by swapping sections 2.13 and 2.14.
Page 4, para 8. It seems as if a different method of in vitro double stranded RNA synthesis was used for the control to what was used for the target genes. Is this correct? If so, is the control comparable?
Page 6. Very nice micrographs.
Page 6 figure legend. Redundant text – “This is a figure”. It is unclear what “Schemes follow the same formatting” means. Is this necessary?
Page 7, paragraph 1. Please state the test name and ideally relevant test statistics as well as P values in the brackets. Please also include the actual P values, rather than a summary (e.g. P < 0.05).
Page 8, para 3. I suggest that the statement is changed from “a well-known PG in mosquitoes16—in the ovary” to “a well known PG, in the ovary of mosquito 16” (if this accurately describes the situation).
Figure 4. Although not essential, graph style could be standardised between panels wherever possible. E.g. column or line chart.
Page 11, paragraph 2. Similar to that mentioned above, it is confusing why in the first sentence COX is mentioned as having a role in the production of PGH2 but COX-like genes were measured in Aedes albopictus. Is this because COX is not present in this species. If so, please state the range of organisms that COX is present. It sounds like this does not include insects. Please state this clearly.
Page 16. Change “disappeared” to “disappear”
Page 16, para 4. The statement “Among PGs, PGE2 was detected in mos-quitoes” suggest this this was detected in the current study. Please re-write to indicate this is prior research. Change “mos-quitoes” to “mosquitoes”.
Page 16, para 4. Italicize “Plasmodium”, “An. Albimanus” and other scientific names.
Fig S1. Please describe in the legend which data are represented by columns, and which data are represented by points and lines.
Author Response
Comment #1-1: Page 1, para 2. It is not clear what is meant by the term “cell-dumping”. Please explain this term at when it is first used.
Response: Explained as follows: “The reproductive role of PGs is extended to mediate oogenesis in a fruit fly, Drosophila melanogaster, wherein PGE2 facilitates a nurse cell-dumping process to growing oocytes, in which nurse cell cytoplasm is dumped into the oocyte [7].”
Comment #1-2: Page 1, para 3. The statement is made that insect genomes do not contain any COX ortholog. However, the second sentence of the paragraph mentions insects and the third paragraph discusses the action of COX. This is confusing as it suggests that COX proteins are found in insects. Furthermore, later in the paragraph states “specific PG syntases convert the COX product..” It is unclear how COX could be playing a role in insects if insects dot contain any COX orthology. Later, in the Discussion, the statement is made that COX proteins occur in vertebrates and not insects. This should be stated here in the Introduction.
Response: Clarified as follows: “In PG biosynthesis, insects use COX-like peroxidases called peroxynectin [14,15] and heme peroxidase [16], which are likely to act in the same way as COX, because insect ge-nomes do not contain any COX ortholog [17].”
Comment #1-3: Page 3, paragaphs 2 and 3. The section on “RT-PCR and RT-qPCR” is highly confusing. The authors refer to “real time PCR” and it seem that they abbreviate this as RT-PCR. However, the abbreviation “RT-PCR” should be used to refer to “Reverse Transcriptase PCR”. It does not seem that the authors mean Reverse-transcriptase PCR in the first sentence of paragraph 2, as the sentence mentions the use of taq polymerase. Revese trancriptase PCR requires the use of a reverse transcriptase, but only taq is mentioned here. Istrongly suggest that the authors do not use the term “Real time PCR” but instead use “quantitative PCR”. It is not clear how the methods described in paragraph 3 differ from those in paragraph 2 (it seems some of the Methods are repeated?).
Response: Corrected into quantitative PCR
Comment #1-4: Page 1, first paragraph. Mosquitoes do not “cause” viruses. I suggest the words “and it is a capable vector causing” are replaced with “including”
Response: Changed into ‘transmitting’
Comment #1-5: Page 1, paragraph 2. Incorrect term “vitellogenesis of fat body”. Should this be “vitellogenesis by the fat body”
Response: Corrected as suggested.
Comment #1-6: Page 1, pare. 2. Change “spermathecal” to “spermathecae”
Response: Corrected as suggested.
Comment #1-7: Page 1, para 2. Change “in a fruit fly” to “in the fruit fly”
Response: Corrected as suggested.
Comment #1-8: Page 2, para 4. What was the final concentration of DMSO in the treatments?
Response: Added as follows: “dimethyl sulfoxide (DMSO, 10%)”
Comment #1-9: Page 3, para 4. Should “sutter” be capitalised? Is this a proper noun?
Response: Yes, it is corrected as suggested.
Comment #1-10: Page 3, paras 5 and 6. The first part of Section 2.7 seems to repeat what is said in Section 2.6.
Response: Deleted in Section 2.7.
Comment #1-11: Page 3, para 7. What was the strain of mouse? State how the mice were anaesthetised.
Response: The strain is written as follows: “from an out-bred mouse line (SLC, Shizuoka, Japan)”
Comment #1-12: Page 4, para 6 (section 2.13). Should “before 24 h bloodmeal” be “24 h before a blood meal”?
Response: Corrected as suggested.
Comment #1-13: Page 4, para 7 (section 2.14). Does this section describe the double stranded RNA constructs that were used in section 2.13? If so, I suggest that this information be placed before section 2.13, possibly by swapping sections 2.13 and 2.14.
Response: Corrected as suggested.
Comment #1-14: Page 4, para 8. It seems as if a different method of in vitro double stranded RNA synthesis was used for the control to what was used for the target genes. Is this correct? If so, is the control comparable?
Response: Clarified as follows: “Template DNA was amplified with gene-specific primers (Table S1) containing T7 promoter sequence (5’-TAATACGACTCACTATAGGGAGA-3’) at the 5’ end. The resulting PCR product was used to in vitro synthesize double-stranded RNA (dsRNA) encoding Ae. albopictus genes using T7 RNA polymerase with NTP mixture at 37ï‚°C for 3 h according to the method outlined by Vatanparast et al. [24] using a MEGAscript RNAi kit (Ambion, Austin, TX, USA). dsRNA was mixed with a transfection reagent Metafectene PRO (Bion-tex, Plannegg, Germany) at a 1:1 (v/v) ratio and incubated at 25ï‚°C for 30 min to form lipo-somes.
In the injection experiment, 300 ng of dsRNA was injected into 4-day-old females (1 day before BF) using a PV830 microinjector under a SZX-ILLK200 stereomicroscope. A control dsRNA (‘dsCON’) was prepared by the same method described above.”
Comment #1-15: Page 6. Very nice micrographs.
Response: Thanks!
Comment #1-16: Page 6 figure legend. Redundant text – “This is a figure”. It is unclear what “Schemes follow the same formatting” means. Is this necessary?
Response: These unnecessary words are deleted.
Comment #1-17: Page 7, paragraph 1. Please state the test name and ideally relevant test statistics as well as P values in the brackets. Please also include the actual P values, rather than a summary (e.g. P < 0.05).
Response: Treatments are rephrased as follows: “Injection of dexamethasone (‘DEX’)—which serves as a general inhibitor of eicosanoid biosynthesis by inhibiting PLA2— impaired egg production. Injection of PG biosynthe-sis inhibitors (aspirin (‘ASP’) or ibuprofen (‘IBU’)) also inhibited egg production, whereas naproxen (‘NAP’), an LT biosynthesis inhibitor, did not (Figure 2A). The addi-tion of PGE2 rescued the oogenesis of females treated with ASP (Figure 2B). The inhibi-tory activity of ASP on oogenesis led to a reduction in the number of laid eggs (Figure 2C).”
Comment #1-18: Page 8, para 3. I suggest that the statement is changed from “a well-known PG in mosquitoes16—in the ovary” to “a well known PG, in the ovary of mosquito 16” (if this accurately describes the situation).
Response: Rephrased as follows: “a well known PG in mosquitoes [16]—in the ovary”
Comment #1-19: Figure 4. Although not essential, graph style could be standardised between panels wherever possible. E.g. column or line chart.
Response: Graphs are revised to be similar size.
Comment #1-20: Page 11, paragraph 2. Similar to that mentioned above, it is confusing why in the first sentence COX is mentioned as having a role in the production of PGH2 but COX-like genes were measured in Aedes albopictus. Is this because COX is not present in this species. If so, please state the range of organisms that COX is present. It sounds like this does not include insects. Please state this clearly.
Response: Insects do not have COX orthologs. Thus COX-like peroxidases behave like mammalian COX. We rephrased the sentences to be clear as follows: “Based on the fact that a specific peroxidase called peroxynectin (Pxt) performs COX-like catalytic activity in insects [20,27], peroxidase genes were annotated from the genome of Ae. albopictus and assessed by a clustering analysis with known COX-like genes in insects (Figure S3). Five peroxidases of Ae. albopictus were clustered with the four known COX-like Pxts (Figure S3A). All these peroxidases possess heme-binding domain (Figure S3B). Comparing these peroxidase genes in terms of their expressions after BF, only two peroxidases (Aa-POX19 and Aa-POX20) were highly inducible to BF (Figure 4E).”
Comment #1-21: Page 16. Change “disappeared” to “disappear”
Response: Corrected as follows: “and then lost”
Comment #1-22: Page 16, para 4. The statement “Among PGs, PGE2 was detected in mos-quitoes” suggest this this was detected in the current study. Please re-write to indicate this is prior research. Change “mos-quitoes” to “mosquitoes”.
Response: Corrected as follows: “Among PGs, PGE2 was detected in mosquitoes because specific stellate cells of Mal-pighian tubules possessed PGE2 in Ae. aegypti to mediate fluid secretion [31].”
Comment #1-23: Page 16, para 4. Italicize “Plasmodium”, “An. Albimanus” and other scientific names.
Response: Corrected as suggested.
Comment #1-24: Fig S1. Please describe in the legend which data are represented by columns, and which data are represented by points and lines.
Response: Corrected as follows: “Bar graphs represent the effect of BF without any ASP treatment. Line graphs represents the effect of different ASP doses.”

Reviewer 2 Report
1. Why the aspirin, dexamethasone, ibuprofen and naproxen were chooosed for injection in this study? How was the injection concentration of each chemical substance determined? In addition, does 10% DMSO has affected the development of Aedes albopictus eggs compared to untreated group?
2. Is the inhibition of prostaglandin E2 biosynthesis by aspirin and ibuprofen specific? Why did you choose aspirin for further study between these two inhibitors under the condition that both aspirin and ibuprofen significantly inhibited egg production?
3. How many female mosquitoes were used for egg counting and egg production experiments?
4. In the experiment in Fig.3.4A, how was the injection volume of JH determined? Is it possible that JH can induce PGE2 production at other concentrations? Why not directly measure the PGE2 content to determine the regulatory effect of 20E/JH on PGE2 production?
5. In the experiment in Fig. 8B, could aspirin injection alone inhibit the expression of vitellogenin? I think this is a necessary supplementary experiment, which can provide support for the hypothesis in Fig. 9.
6. The effect of PGE2 synthesis pathway gene silencing on egg development should be combined with the results of part 3.3 (PG biosynthetic genes of Ae. albopictus). In addition, how did gene silencing in the PGE2 synthesis pathway affect nurse cell?
7. The results of 20E biosynthetic gene silencing inhibiting oogenesis of Ae. Albopictus should be integrated with the results of part 3.4 (PGE2 production is controlled by 20E after BF).
Author Response
Comment #2-1: Why the aspirin, dexamethasone, ibuprofen and naproxen were chooosed for injection in this study? How was the injection concentration of each chemical substance determined? In addition, does 10% DMSO has affected the development of Aedes albopictus eggs compared to untreated group?
Response: As mentioned in the text, these chemicals are specific inhibitors in different steps of PGE2 biosynthesis. 10% DMSO did not give any adverse effect on the mosquito development of Ae. albopictus.
Comment #2-2: Is the inhibition of prostaglandin E2 biosynthesis by aspirin and ibuprofen specific? Why did you choose aspirin for further study between these two inhibitors under the condition that both aspirin and ibuprofen significantly inhibited egg production?
Response: Both inhibitors are specific to COX and prevent PG biosynthesis. Either inhibitor might be OK to test our hypothesis. However, ASP was slightly more potent than IBU though they were not significantly different.
Comment #2-3: How many female mosquitoes were used for egg counting and egg production experiments?
Response: As mentioned in section 2.7, 10 females were used for each treatment.
Comment #2-4: In the experiment in Fig.3.4A, how was the injection volume of JH determined? Is it possible that JH can induce PGE2 production at other concentrations? Why not directly measure the PGE2 content to determine the regulatory effect of 20E/JH on PGE2 production?
Response: As mentioned in section 2.9, 100 nL was injected. In this experiment, our hypothesis was that 20E stimulate PG biosynthesis by stimulating its biosynthetic pathway. Thus, we measured the expression level of PLA2, which catalyzes the committed step of the biosynthesis.
Comment #2-5: In the experiment in Fig. 8B, could aspirin injection alone inhibit the expression of vitellogenin? I think this is a necessary supplementary experiment, which can provide support for the hypothesis in Fig. 9.
Response: As per Fig. 8, ASP did not inhibit vitellogenin synthesis, which is controlled by 20E.
Comment #2-6: The effect of PGE2 synthesis pathway gene silencing on egg development should be combined with the results of part 3.3 (PG biosynthetic genes of Ae. albopictus). In addition, how did gene silencing in the PGE2 synthesis pathway affect nurse cell?
Response: I believe the reviewer was asking about Fig. 7, which shows the RNAi effect on egg development. This result demonstrates the crucial roles of the main three biosynthetic steps of PGE2 in the mosquito reproduction. The influence of PGE2 on nurse cell dumping is explained in Discussion as follows: “Nurse cell dumping is required for oocyte development in insects containing poly-trophic ovarioles [32]. In Drosophila, such nurse cell dumping is mediated by PGs via cytoskeleton rearrangement by bundling actin filament through Fascin [7]. In S. exigua, PGE2 stimulates a small GTPase, Cdc42, to activate Fascin by binding to its specific PGE2 receptor [33]. Later, the PGE2 binding to its receptor has been shown to trigger cAMP release and the subsequent up-regulation of Ca2+, which activates small G pro-teins to stimulate actin polymerization and bundling to modulate cytoskeletal rear-rangement [34]. The inhibition of PG biosynthesis prevented the oogenesis of S. exigua, because nurse cell dumping did not occur, as was the case in our current study using mosquitoes [16]. This suggests that PGE2 induces Ca2+ signal, which activates the cyto-skeletal rearrangement of nurse cells to dump their contents into the growing oocytes in Ae. albopictus.”
Comment #2-7: The results of 20E biosynthetic gene silencing inhibiting oogenesis of Ae. Albopictus should be integrated with the results of part 3.4 (PGE2 production is controlled by 20E after BF).
Response: RNAi specific to Shade, which catalyzes E to form 20E, is combined in Fig. 6. This figure shows the physiological role of the biosynthetic gene product in the regulation of PGE2 biosynthesis. All these results are in Section 3.4.

Round 2
Reviewer 2 Report
This manuscript should be accepted.